# Prophylactic Central Neck Lymph Node Dissection Adds No Short-Term Benefit to Total Thyroidectomy for Differentiated Thyroid Cancer

**DOI:** 10.3390/medicina59020239

**Published:** 2023-01-27

**Authors:** Rosen Dimov, Gancho Kostov, Mladen Doykov, Luboslav Dimov, Boyan Nonchev, Rositsa Dimova, Bozhidar Hristov

**Affiliations:** 1Department of Special Surgery, Medical Faculty, Medical University of Plovdiv, 4001 Plovdiv, Bulgaria; 2Department of Surgery, University Hospital “Kaspela”, 4001 Plovdiv, Bulgaria; 3Department of Urology and General Medicine, Medical Faculty, Medical University of Plovdiv, 4001 Plodiv, Bulgaria; 4Department of Urology, University Hospital “Kaspela”, 4001 Plovdiv, Bulgaria; 5Department of Endocrinology, Medical Faculty, Medical University of Plovdiv, 4001 Plovdiv, Bulgaria; 6Department of Endocrinology, University Hospital “Kaspela”, 4001 Plovdiv, Bulgaria; 7Department of Health Management and Health Economics, Faculty of Public Health, Medical University of Plovdiv, 4001 Plovdiv, Bulgaria; 8Second Department of Internal Diseases, Section of Gastroenterology, Medical Faculty, Medical University of Plovdiv, 4001 Plovdiv, Bulgaria; 9Department of Gastroenterology, University Hospital “Kaspela”, 4001 Plovdiv, Bulgaria

**Keywords:** differentiated thyroid cancer, prophylactic central neck lymph node dissection

## Abstract

*Background and Objectives*: To answer the research question: “Is prophylactic central neck lymph node dissection (pCNLD) beneficial among differentiated thyroid carcinoma (DTC) patients?” *Materials and Methods*: This was a retrospective cohort study enrolling DTC patients treated at the University Hospital Kaspela, Bulgaria, from 30 January 2019 to October 2021. The predictor variable was presence of pCNLD (total thyroidectomy with vs. without pCNLD). The main outcome variables were postoperative complications (i.e., vocal cord paralysis, hypoparathyroidism, postoperative bleeding, and adjacent organ injury) and recurrence parameters. Appropriate statistics were computed with the significant level at *p* ≤ 0.05. *Results*: During the study period, 300 DTC patients (59.7% with pCNLD; 79.3% females) with an average age of 52 ± 2.8 years were treated. The mean follow-up period of the entire cohort was 45.8 ± 19.1 months. On bivariate analyses, TT with pCNLD, when compared to TT alone, required longer surgical time (mean difference: 9.4 min), caused nearly similar complications (except transient hypothyroidism: *p* = 0.04; relative risk, 1.32; 95% confidence interval, 1.0 to 1.73), and no significantly different recurrence events, time to recurrence, and recurrent sites. The benefit–risk analyses using the number needed to treat and to harm (NNT; NNH) also confirmed that TT plus pCNLD was not very beneficial in DTC management. *Conclusion*: The results of this study refute the benefit of pCNLD in DTC patient care with TT. Further well-designed studies in a larger cohort with a longer follow-up period are required to confirm this conclusion.

## 1. Introduction

Differentiated thyroid cancer (DTC) is the most common malignant tumor of the thyroid gland [1]. Many studies and meta-analyses have shown recurrence in about 30% of cases. Approximately 40% of these patients develop metastatic cervical lymph nodes at the time of diagnosis. Furthermore, micro-metastases in clinically negative lymph nodes can be detected in 60–80% of the cases [2,3]. However, the benefit–risk of prophylactic central node dissection (pCNLD) during initial surgery in DTC patients remains controversial [4,5,6]. The aim of this study is to answer the research question: “Is pCNLD coupled with total thyroidectomy (TT) beneficial among DTC patients?” The investigators hypothesized that pCNLD would decrease tumor recurrence and complications in DTC patients undergoing TT. The specific aims were (1) to conduct a retrospective cohort study, (2) to identify DTC patients undergoing TT with versus without pCNLD, (3) to assess treatment outcomes between the two patient groups, and (4) to develop the recommendation on the necessity of pCNLD during TT for DTC.

## 2. Materials and Methods

### 2.1. Study Design and Sample Description

This was a retrospective cohort analysis of data of DTC patients receiving TT with and without pCNLD at the Department of General Surgery, University Hospital Kaspela-Plovdiv, Bulgaria, from January 2019 to October 2021 by the same surgeon (R.D.). The Institutional Review Board approved the project (IRB No. 28 of 2018-12-03). Before the treatment, all patients signed a written consent form for treatment and research, in compliance with the ethical principles specified in the World Medical Association Declaration of Helsinki. Patients were included if they (1) received TT with or without pCNLD, (2) had pre-operative diagnosis of PTC by fine-needle aspiration biopsy of the primary tumor and ultrasound of cervical lymph nodes, (3) were aged ≥18 years, and (4) had DTC with clinical stage T1 to T4 without extrathyroidal extension and no clinical evidence of lateral lymph node involvement before the surgery. The exclusion criteria were patients (1) with diagnosis of a histological type other than DTC, (2) with mixed tumors, (3) with inadequate data documentation, and (4) who had undergone lymph node dissection of lateral compartments (therapeutic CNLD). 

### 2.2. Study Variables

The predictor variable was surgical technique, i.e., TT with versus without pCNLD, which were recorded as a binary parameter. In the TT-alone group, the patients were totally thyroidectomized due to the incidental finding of the cancer after the final pathological report with the patient’s willingness (consent). The main outcome variables included: (1) complications, that is, recurrent laryngeal nerve (RLN) palsy, temporary and permanent hypoparathyroidism, postoperative bleeding, injury to adjacent organ (esophagus, trachea, etc.); (2) recurrence parameters. The vocal cord status was assessed by routine pre- and postoperative indirect laryngoscopy and ultrasound. Temporary vocal cord palsy was defined as decreased or absent vocal cord mobility that resolved within 6 months after surgery. Permanent vocal cord palsy was defined as vocal cord dysfunction persisting more than 6 months after the initial surgery. To determine hypoparathyroidism, serum calcium levels were investigated preoperatively and on each postoperative day. Parathyroid hormone (PTH) levels were tested daily (normal level: 12–88 pg/mL). Symptomatic hypocalcemia was diagnosed if the calcium level was <2.17 mmol/L and patients complained of any symptoms (such as perioral and digital paresthesia, tetany, and palpitation), irrespective of the duration of hospital stay. Routine calcium supplementation was given to all patients. Permanent hypocalcemia or hypoparathyroidism was defined as an ongoing need for calcium or vitamin D supplementation >12 months.

The other study parameters were (1) demographic (age and gender) and (2) tumor-related (tumor staging, TNM status, and histopathological results). Age was recorded as a continuous parameter, gender was categorized into binary, and the tumor status was assessed as nominal scales.

### 2.3. Data Management and Statistical Analysis

The data were collected using Microsoft Excel 365 and analyzed using the statistical software MedCalc 20. Continuously measured and normally distributed variables were described through their mean values and standard deviations, and between-group comparisons were performed using the t-test for independent samples or one-way analysis of variance (ANOVA). Binary and categorical data were presented in frequencies and percentages, and associations were established through the chi-square test and Fisher’s exact test. All statistical tests were two-tailed and performed using the significant level at *p* < 0.05. Further statistics such as relative risk (RR) and 95% confidence interval (CI) were computed as appropriate.

## 3. Results

During the study period, 300 DTC patients (59.7% with pCNLD; 79.3% females) with an average age of 52 ± 2.8 years were treated. Both patient groups were statistically comparable with regard to demographic and tumor-related parameters, except age. Papillary thyroid carcinoma (PTC), followed by the follicular type, in early stages was the most common histopathological type in both groups. Surgical time was significantly lengthened in the TT plus pCNLD group with the mean difference of 9.4 min (Table 1).

Contralateral cervical node metastatic spreads were found in 15 (13.9%), 5 (31.3%), 29 (60.4%), and 6 (85.7%) of patients with stage T1–T4, respectively. The prelaryngeal (Delphian) lymph node was affected by metastatic processes in 22 (12.3%) cases with prophylactic dissection, with a significant OR of 13.23 (95%CI: 1.97 to 90.101) (Table 2).

Based on the benefit–risk analyses (Table 3), one of every 93 patients would benefit from pCNLD in terms of the absence of transient vocal cord paralysis (when compared to TT alone), while one in every 8 and 165 patients would be harmed by transient and permanent hypoparathyroidism after TT with pCNLD, respectively. By the “rule of thumb”, an NNT <10 is ordinarily considered clinically meaningful because a treatment difference would be routinely encountered in day-to-day clinical practice, and the high NNH is preferred because the higher the NNH, the less important the difference between the two interventions with respect to the adverse outcome of interest [7]. This means that TT plus pCNLD was not very beneficial in DTC management.

Concerning recurrence of the disease, there was no statistically significant difference in recurrence events, time to recurrence, and recurrence sites after the average follow-up period of 45.8 ± 19.1 months (Table 4).

In all the patients with advanced cancer (T3–T4), a poor histological variant was diagnosed. The lack of metastases in the central neck compartment during prophylactic dissections did not reduce the risk of recurrence (*p* = 0.884) (Table 5).

## 4. Discussion

The latest American Thyroid Association (ATA) Guidelines recommend that pCNLD be performed in patients with DTC and clinically uninvolved central neck lymph nodes only for bulky advanced tumors (i.e., T3–T4). However, the use of pCNLD remains controversial, when preoperative cervical lymph node involvement is absent (i.e., cN0) [8]. It is generally accepted that lymph node metastasis amid histopathological tumor types is usually and highly associated with tumor recurrence. Many surgeon-authors suggested TT with pCNLD for cN0 DTC to obtain adequate tumor staging, reduce the recurrence risk, and improve the patient’s overall survival [6,9]. On the contrary, extensive surgery is commonly linked to complications. The European Society of Medical Oncology (ESMO), British Thyroid Association (BTA), and US National Comprehensive Cancer Network (NCNN) have promoted CNLD only in cases of positive lymph nodes (hence, in this case, the CNLD should be called “therapeutic CNLD”) [10,11,12,13]. However, all of these recommendations are classified as weak, with low-quality evidence based on information from retrospective nonrandomized studies. 

To solve this controversy, the investigators of this study designed and implemented a retrospective cohort study on treatment outcomes in 300 DTC patients undergoing TT with and without pCNLD. The study’s hypothesis was that pCNLD would decrease tumor recurrence and complications in DTC patients undergoing TT. Our main results rejected the hypothesis. Despite the significantly higher patient age (*p* = 0.03; 95% CI, −1.35 to −0.05), bivariate analyses demonstrated nearly similar complications (except transient hypothyroidism: *p* = 0.04; RR, 1.32; 95% CI, 1.0 to 1.73), and no significantly different recurrence events, time to recurrence, and recurrent sites. The benefit–risk analyses using NNT and NNH also confirmed that TT plus pCNLD was not very beneficial in DTC management.

Roh et al. [14] analyzed treatment outcomes in a cohort of 155 PTC patients, 40 (or 25.8%) of whom underwent TT with pCNLD in spite of cN0. They found that patients undergoing TT with neck node dissection experienced greater overall morbidity, compared with those receiving TT alone (50% vs. 12.3%; *p* < 0.001). In our study, patients with TT alone had 53 complication events (i.e., 4 vocal cord paralysis, 46 hypoparathyroidism (albeit transient both), 1 postoperative bleeding, and 2 adjacent organ injuries), while patients with TT and pCNLD developed 97 postoperative adverse events (that is, 4 transient vocal cord paralysis, 90 transient hypoparathyroidism, 1 postoperative bleeding, and 2 neighboring organ injuries) (RR, 1.15; 95% CI, 0.89 to 1.47; *p* = 0.28).

The difference between our findings and those reported by Roh et al. [14] may be due to the fact that we had significantly higher rates of transient hypoparathyroidism in both patient groups and/or that Roh et al. [14] included only PTC from Republic of Korea. The effect of the tumor’s histological types and patient’s ethnicity (or genetic effect) on complication events after TT with and without pCNLD remains unknown and requires further investigations. However, as proposed by Roh et al. [14], transient hypoparathyroidism appears to result from the extension of nodal dissection to include the central neck to lateral cervical compartment, which increases vascular compromise in the dissected central neck and parathyroid glands. Careful preservation of parathyroid vascularity and/or autotransplantation of devascularized parathyroid glands could help reduce postoperative hypoparathyroidism in thyroid cancer patients involving macroscopic nodal metastasis where aggressive neck management is required. 

Another particular concern is metastasis to clinically and radiologically negative neck nodes in DTC patients. To the best of our knowledge, three recent systematic reviews of randomized trials with meta-analysis had demonstrated the absence of benefit of pCNLD in thyroid cancer patients with cN0 undergoing TT (despite the fact that central lymph node metastasis CLNM may be as high as 40% in this patient group and has been found to be associated with poor prognosis) [15,16,17].

Among those meta-analytic publications, Wen et al. [15] depicted 4 clinicopathological predictors of central lymph node metastasis in cN0 PTC patients, indicating the necessity of pCNLD. These predictors included (1) male patient gender (vs. female gender: odds ratio OR, 2.43; 95% CI, 2.43 to 3.22; *p* < 0.001), (2) tumor multifocality (vs. unifocality: OR, 1.88; 95% CI, 1.54 to 2.29; *p* < 0.001), (3) tumor size > 5 mm (vs. ≤5 mm: IR, 1.84; 95% CI, 1.55 to 2.18; *p* < 0.001), and (4) extrathyroidal extension (ETE) of the tumor (vs. other: OR, 2.38; 95% CI, 1.65 to 3.44; *p* < 0.001).

Sanabria et al. [16] met the conclusion that low-risk PTC, which was defined as intrathyroidal tumors without macroscopic ETE, did not require pCNLD (recurrences in TT-alone group vs. in TT + pCNLD: 9/354 or 2.5% vs. 11/409 or 2.7%; risk difference (RD), 0; 95% CI, −2% to 2%; *p* = 0.98; NNT, 500), while the rate of permanent hypoparathyroidism was higher in the TT + pCNLD group (RD, 3%; 95% CI, 0% to 6%; *p* = 0.02; NNH = 33.3). Our results support those of Sanabria et al. [16] because we found the recurrence rate of 2.5% and 2.2% in patients undergoing TT without and with pCNLD (*p* = 0.89; RR, 0.9 (95% CI, 0.21 to 3.96); NNT, 408.66 (95% CI, 30.89 harm to ∞ to 26.83 benefit. However, these results of our study should be interpreted cautiously because of the low incidence of recurrences (post hoc power, 3.6%). In other words, we would have needed 60,333 patients in each treatment arm (total *n* = 120,666), or 15,457.2 months (over 1200 years) to detect the statistical significance of tumor recurrences between the patient groups with the study power of 80% at α = 0.05.

One retrospective study from Italy (*n* = 1087) reported that concomitant complete contralateral paratracheal lymph node dissection should be performed only in cases of positive intraoperative frozen section results (or alternatively, by experienced surgeons and/or in the high hospital-volume setting in order to counterbalance between benefit of tumor recurrence reduction and postoperative complications) [18]. Although micrometastases can escape from the frozen section examination, several recently published studies suggested the usefulness of this intraoperative investigation [19,20]. For example, in a South Korean study (*n* = 216), the sensitivity, specificity, positive predictive value (PPV), and negative predictive value (NPV) of the frozen analyses of central lymph nodes in PTC patients were 94.6% (88/93), 100% (123/123), 100% (88/88), and 96.1% (123/128), respectively. Statistical significance was reached, and the diagnostic agreement rate between the frozen analyses and the permanent pathologic report was very high (κ index = 0.962). The likelihood ratio was 23.57 (*p* < 0.001) [19]. These findings confirmed the results of a meta-analysis paper published in 2019 [20].

Even though the relatively high hospital volume (*n* = 300 in 31 months, or approximately 10 patients per month operated by a single surgeon) seems to elevate this study’s strength, we recognize some study limitations. First, this study suffers from its retrospective study design, a low sample size, and the short follow-up period. As mentioned above, in order to detect the statistical significance in relation to tumor recurrences between the TT-alone group and the TT + pCNLD group, over 1200 years would have been required from our Bulgarian medical university. Second, we cannot conclude whether the differences on patients’ ages and surgical time between the two treatment arms could affect treatment outcomes in this study. Further well-designed studies in a large cohort are essentially important to confirm our results. Last but not least, the study’s external validity (generalizability) cannot be estimated, i.e., we do not know whether our treatment outcomes, which were based on the single surgeon, are able to be applied to those of other surgeons in other hospitals and world regions.

## 5. Conclusions

Our findings suggest that TT with pCLND in DTC patients has neither statistically significant advantages nor complications during the relatively short observation period. Based on these results, pCLND appears to be unnecessary for this patient group.

## Figures and Tables

**Table 1 medicina-59-00239-t001:** Demographic, tumor-related parameters and predictor variable.

Parameters	TT Alone (*n* = 121)	TT Plus pCNLD (*n* = 179)	*p*-Value (RR; 95% CI)
Age	51.6 ± 3.4	52.3 ± 2.3	**0.03 (N/A; −35 to −0.05)**
Female gender	99 (81.8)	141 (78.8)	0.31 (1.2; 1.06 to 1.35)
Tumor size (cm^3^)	1.7 ± 1.5	1.8 ± 1.4	0.56 (N/A; −0.43 to 0.23)
Surgery time (min)	73.2 ± 22.4	82.6 ± 18.8	**0.0001 (N/A; −14.11 to −4.69)**
Stage			
I	97 (80.2)	143 (79.9)	0.53 (0.25; 0.18 to 0.34)
II	24 (19.8)	36 (20.1)	
T stage			
T1	71 (58.7)	108 (60.3)	
T2	12 (9.9)	16 (8.9)	1.0 (N/A)
T3	31 (25.6)	48 (26.8)	
T4	7 (5.8)	7 (3.9)	
N stage			
N0	0	104 (58.1)	N/A
N1a	0	64 (35.8)	
N1b	0	11 (6.1)	
Histological types			
Papillary thyroid carcinoma	67 (55.4)	134 (74.9)	
Oncocytic carcinoma of thyroid	8 (6.6)	4 (2.2)	0.70 (N/A)
Hyalinizing trabecular tumor	4 (3.3)	1 (0.6)	

Note: Continuous measurements are listed as mean ± standard deviation (range); categorical measurements are listed as number (percentage). RR—relative risk; CI—confidence interval; N/A—not available. Statistically significant *p* values are indicated in bold typeface. pCNLD—prophylactic central neck lymph node dissection.

**Table 2 medicina-59-00239-t002:** Analysis of metastasis of cervical neck lymph nodes.

Tumor Staging	Ipsilateral Neck Node Metastasis	Contralateral Neck Node Metastasis	Prelaryngeal Node Metastasis	*p*-Value (RR, 95% CI)
T1	39(36.1)	15 (13.9)	3(2.8)	
T2	12(75.0)	5 (31.3)	3(18.7)	
T3	44(91.7)	29 (60.4)	11(22.9)	
T4	7(100)	6 (85.7)	5(71.4)	

**Table 3 medicina-59-00239-t003:** Complications grouped by the primary predictor variable.

Complications	TT Alone (*n* = 121)	TT Plus pCNLD (*n* = 179)	*p*-Value (RR; 95% CI)	NNT or NNH
Vocal cord paralysis				
Transient	4 (3.4)	4 (2.2)	0.57 (0.68; 0.17 to 2.65)	93.36 (NNT)
Permanent	0	0	N/A	N/A
Hypoparathyroidism				
Transient	46 (38.0)	90 (50.3)	**0.04 (1.32; 1.0 to 1.73)**	8.16 (NNH)
Permanent	4 (3.3)	7 (3.9)	0.78 (1.18; 0.35 to 3.95)	165.33 (NNH)
Postoperative bleeding	1 (0.8)	1 (0.6)	0.78 (0.68; 0.04 to 10.7)	373.43 (NNT)
Adjacent organ injury	2 (1.7)	2 (1.2)	0.69 (0.68; 0.1 to 4.73)	186.72 (NNT)

Note: Continuous measurements are listed as mean ± standard deviation (range); categorical measurements are listed as number (percentage). RR—relative risk; CI—confidence interval; NNT—number needed to treat; NNH—number needed to harm; N/A—not available. Statistically significant *p* values are indicated in bold typeface.

**Table 4 medicina-59-00239-t004:** Recurrence in the study samples grouped by the primary predictor variable.

Variables	TT Alone (*n* = 121)	TT with pCNLD (*n* = 179)	*p*-Value (RR; 95% CI)
Follow-up period (month)	45.1 ± 19.9	46.2 ± 18.5	0.62 (N/A; −5.52 to 3.32)
Recurrence	3 (2.5)	4 (2.2)	0.89 (0.9; 0.21 to 3.96)
Time to recurrence (month)	29.5 ± 20.5	32.4 ± 23.1	0.27 (−8.02 to 2.22)
Recurrence Sites			
Central lymph node(s)	3 (2.5)	0	1.0 (N/A)
Lateral lymph node(s)	0	3 (1.7)	
Thyroid bed	0	1 (0.6)	

Note: Continuous measurements are listed as mean ± standard deviation (range); categorical measurements are listed as number (percentage). RR—relative risk; CI—confidence interval; N/A—not available. Statistically significant *p* values are indicated in bold typeface.

**Table 5 medicina-59-00239-t005:** Positive lymph nodes connected with diseases recurrence.

	Sig.	df	*p*-Value	Mean Difference	Std. Error Difference	95% Confidence Interval	
						Lower	Upper
TT + pCNLD	0.883	218	0.884	−0.037	0.253	0.535	0.761
TT		3.084		−0.037	0.291	0.448	0.874

## Data Availability

The data presented in this study are available on request from the corresponding author. The data are not publicly available, due to institutional restrictions.

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
