# Peer review of "Prophylactic Central Neck Lymph Node Dissection Adds No Short-Term Benefit to Total Thyroidectomy for Differentiated Thyroid Cancer"

_medicina, 2023, doi:10.3390/medicina59020239_

Round 1

Reviewer 1 Report (Previous Reviewer 3)

Even though the authors claim to have excluded the incidentally found thyroid cancers from thes study, and that all patients had FNA proven thyroid cancer, in the Methods section they state that:

The patients in our study were separated in two groups with thyroidectomy plus prophylactic central neck lymph node dissection(after FNA and ultrasound diagnosed cancer), and with thyroidectomy alone(in this group patients were with thyroidectomy due to incidental finding of the cancer after final pathological report and due to the patients insistence.

Please clarify.

2. In Table 1, it seems that there is a high prevalence of tall cell type tumors 55.3% in TT and 74.8% in TT+CLND. Please verify the data, because this can't be true. Also, in the same table, you have another tall cell type line...

3. I feel that you should include the original population who underwent TT and was found to have T4 disease with extrathyroidal extension post-opertaively, excluding the data from the therapeutic CLND. 

4. Please add a clear statement on the limitations of the study, including the short duration of follow up and the relatively small number of patients in each group.

5. Please add a clear statemment on the higher incidence of transient hypoparathyroidism in patients treated with CLND, as compared to TT alone.

6. Please, have a native English speaker re-evaluate the manuscript.

Author Response

Thank you very much for your editing! We made corecctions following your recommendations. I hope now our manuscript is suitible for publication.

Reviewer 2 Report (New Reviewer)

This is a really good article. The topic is of high relevance, especially cause this situation with lymph node dissection persists in other cancer types too. For example by endometrial carcinoma.

So I see no major issues but only some minors:

1) Table 2 is not clear. What is the reference group for ORs. For example, if Prelaryngeal has OR of 13, what is here the reference group, Prelaryngeal compared with?

2) Table 3; you write p=0.00. I would rather write <0.001 as this would be clearer for readers

3) "A strength of the study is that 267
all cases were operated on by the same experienced surgical team increases the reliability 268
of the results by excluding differences between surgeons.".  However, this is also a limitation. This may be the case that by this team no metastases occur when no
lymph node dissection was performed cause doctors removed the primary tumor very well; however, by another less experience team, this could be not the case.  I would say, single center study rather than multi-center study is a limitation.

Author Response

Thank you very much for your editing! We made corecctions following your recommendations. 

This manuscript is a resubmission of an earlier submission. The following is a list of the peer review reports and author responses from that submission.

Round 1

Reviewer 1 Report

The study compare the complications rate and benefits of prophylactic central neck lymph node dissection to thyroidectomy alone in differentiated thyroid cancer. 

However, there are some concerns about the study:

1. In Materials and methods: Why the authors eliminate the extrathyroidal extension?

2. Why the authors eliminate children from the study (<18 years of age)?

3. The authors should declare that the patients was diagnosed with cN0 status. 

4. Results: Table 2. 

N2 is not defined in TNM staging (AJCC 8th)

Author Response

Thank you very much for your editing! I think our article look much better. 

Thank you!

  1. In Materials and methods: Why the authors eliminate the extrathyroidal extension- These patients have an advanced disease, with an increased risk of metastatic lesions in the central and lateral cervical compartments often required therapeutic dissections.

  2. Why the authors eliminate children from the study (<18 years of age)? Because of national health system regulations patients under 18 are operated by pediatric surgeons.
  3. The authors should declare that the patients was diagnosed with cN0 status.- Patients with prophylactic central neck lymph node dissection always have a proven N0 status, this is achieved with the help of imaging studies, most often ultrasound, the absence of enlarged, suspicious lymph nodes from the central cervical compartment classifies patients in this group 
    4. Results: Table 2. 

    N2 is not defined in TNM staging (AJCC 8th). -Thank you, we correct it, it does not affect the results in this case.

Reviewer 2 Report

The role of prophylactic Central Node Dissection (CND) performed during the initial surgery is  a very interesting and current topic, at the center of scientific debate. However, the paper has some limitations: retrospective case studies (a prospective series would have been much more interesting); a limited follow-up that does not allow to evaluate the role of RAI therapy (when necessary) and the not particularly impressive case series. A number of thyroid surgeons recommend pCND along with TT in cN0 patients to avoid possible local recurrences and to obtain adequate staging to assess the need for RAI therapy.

In DTC Nx some recent works also suggest the possibility of performing a lymphadenectomy of the ipsilateral central hemicompartment. This would limit complications of larger dissections, allowing for more precise staging, useful for successful treatments.

The bibliography is not updated, there are some inaccuracies and there is even a repetition (Rosato's work is cited twice, n.9 and n.26)

Editing needs to be improved. There are spelling errors (Hurtel for Hurtle, terapeutic for therapeutic, metstasis...) spaces and lowercase and uppercase letters not respected.

Author Response

Thank you for your edit! Thanks to your notes, I think our article has better view.

Thank you once again!

The bibliography is not updated, there are some inaccuracies and there is even a repetition (Rosato's work is cited twice, n.9 and n.26) -it is technical mistake, we fixed it. Thank you!

Editing needs to be improved. There are spelling errors (Hurtel for Hurtle, terapeutic for therapeutic, metstasis...) spaces and lowercase and uppercase letters not respected-we think we fixed them. Thank you!

Reviewer 3 Report

The authors did a pretty good job reviewing data from 300 consecutive thyroid surgeries performed for thyroid cancer in their institution. The analysis of these data revealed that both surgical procedures are safe and well tolerated by the patient population.

This data-set should change in a few ways to provide a better understanding of the patient population operated on

1. There should be a note of the number of patients operated for a preoperatively diagnosis of thyroid cancer in each group and what was the number of incidentally discovered thyroid cancers in each subgroup.

2. We should have statistics of the stage of the disease according to the procedure performed, not the other way around.

3. Why were patients with extrathyroidal extension excluded from the analysis?

4. Since the time frame of the study is too small, it should be prudent to avoid conclusions on the efficacy of each strategy to reduce recurrence.

5. The same holds true regarding the small number of cases of recurrence in each group.

6. Since the transient hypoparathyroidism was more common in the CLND group, how can we conclude that both procedures are equally well tolerated?

7. Are there data on permanent hypoparathyroidism in these two cohorts?

8. Editing by a native speaker of Endlish language is needed.

Author Response

Thank you for your edit! We have corrected the article and thanks to your editing it now looks much better.

  1. There should be a note of the number of patients operated for a preoperatively diagnosis of thyroid cancer in each group and what was the number of incidentally discovered thyroid cancers in each subgroup.- The patients in this study were with pre-operatively proven thyroid cancer by FNA, we did not include the accidentally discovered cases.
  2. We should have statistics of the stage of the disease according to the procedure performed, not the other way around.- of course you are right, we consider the possible correlation between the type of surgical procedure according to the stage of the disease
  3. Why were patients with extrathyroidal extension excluded from the analysis? These patients have an advanced T4 stage, with an increased risk of metastatic lesions in the central and lateral cervical compartments often required therapeutic dissections.
  4. Since the time frame of the study is too small, it should be prudent to avoid conclusions on the efficacy of each strategy to reduce recurrence - of course you are right, that’s why we add Considering the short period of observation the present study….
  5. The same holds true regarding the small number of cases of recurrence in each group- you are right like your previous note
  6. Since the transient hypoparathyroidism was more common in the CLND group, how can we conclude that both procedures are equally well tolerated? Transient hypoparathyroidism is treated well by temporary replacement therapy in the patients, usually like our study(we add statistics for permanent-thanks to you) differences disappears when comparing the permanent damage to the Parathyroids
  7. Are there data on permanent hypoparathyroidism in these two cohorts?-We added it, Thank you!
  8. Editing by a native speaker of Endlish language is needed. It is done, Thank you!